# Unassisted self-healing photocatalysts based on Le Chatelier's principle
Aito Takeuchi[1], Yoshitaka Kumabe [1,2] & Takashi Tachikawa [1,2] ✉

Self-healing is a fundamental ability inherent in humans, plants, and other living organisms. To date, a variety of materials with self-healing properties have been developed. However, these materials usually require external inputs such as electric potentials or healing agents to initiate or promote self-healing reactions. Herein, we present a novel self-healing mechanism that operates without any external input, utilizing the dynamic equilibrium between the solid-state and dissolved materials. We employed organic–inorganic perovskites to validate our strategy. Single-particle spectroscopy and imaging demonstrated the spontaneous self-healing of perovskites after photodamage under dynamic equilibrium conditions. Furthermore, we found that perovskites can generate hydrogen in both healed and damaged states. Remarkably, the perovskites exhibited hydrogen generation over four cycles of photodamage and self-healing. The proposed concept and experimental results provide valuable insights for the development of energy conversion and storage systems with improved long-term durability.

The ability to heal is crucial for living organisms. When skin is injured or a bone is fractured, the damaged biological tissues naturally repair themselves over time. Providing self-healing capabilities to other non-biological materials can significantly enhance their durability. For several decades, researchers have been actively investigating biomimetic self-healing materials, as conceptually summarized in Fig. 1a [1,2]. In conventional materials (Fig. 1a, left), self-repairing systems can be classified into those with intrinsic and extrinsic mechanisms. The intrinsic mechanism utilizes the atoms and bonds of the material itself, which is mainly found in the self-healing of polymers [3–5]. The extrinsic mechanism employs healing agents, such as capsules or nanoparticles [6,7]. Overall, these intrinsic and extrinsic self-healing materials require external inputs such as heat, adhesives, or healing agents to facilitate the processes.

Self-healing (photo)catalysts have also been investigated (Fig. 1a, middle) [8–10]. Nocera et al. reported oxygen-generating self-healing photoelectrodes that required an appropriate applied bias [11–13]. In this system, $Co^{4+}$ ions loaded onto the electrode served as catalytic sites. As oxygen generation proceeds via water oxidation, $Co^{4+}$ is reduced, leading to the dissociation of Co cations from the electrode and a decrease in catalytic durability. When the applied potential was sufficient to oxidize the Co cations, the redeposition of Co cations proceeded simultaneously, allowing for further photocatalytic reactions. Other metal-based self-healing photoelectrodes have been reported [14]; however, most of these catalysts cannot self-heal without an external input (e.g., an electric potential) in catalytically active environments.

Herein, we propose a newly designed unassisted self-healing mechanism based on dynamic equilibrium (Fig. 1a, right). When materials in this dynamic equilibrium state are damaged, the equilibrium shifts towards self-healing reactions because of Le Chatelier's principle. This principle states that if a system in equilibrium is disturbed, the system adjusts to minimize the disturbance. Figure 1b illustrates a reaction scheme facilitated by Le Chatelier's principle. In the first step, the original materials reach a dynamic equilibrium between the solid state and their dissolved form. When the materials are damaged or several components are eluted, this dynamic equilibrium is disturbed by the increased concentration of chemicals in the solution. According to Le Chatelier's principle, the equilibrium shifts towards reactions that reduce the concentration of chemicals in the solution (second step). Finally, self-healing of the materials is completed (third step). This self-healing reaction operates without the need for external assistance, and the key factor is maintaining the reaction system in a dynamic equilibrium state. The degradation and self-healing of materials are reversible reactions that allow continued use.

In this study, organic–inorganic perovskites are employed as exemplary materials to demonstrate our concept. Halide perovskites are promising light-harvesting and light-emitting materials for applications such as solar cells, light-emitting diodes (LEDs), and lasers owing to their favorable optical properties [15–20]. However, these perovskites are inherently unstable under humid conditions. Nonetheless, recent studies have demonstrated that perovskites can exist stably in saturated aqueous solutions, reaching a dynamic equilibrium between the solid-state and dissolved phases [21]. Under

[1]Department of Chemistry, Graduate School of Science, Kobe University, 1-1 Rokkodai-cho, Kobe, 657-8501, Japan. [2]Molecular Photoscience Research Center, Kobe University, 1-1 Rokkodai-cho, Kobe, 657-8501, Japan. ✉e-mail: tachikawa@port.kobe-u.ac.jp

**Fig. 1 | Overview of self-healing reactions based on Le Chatelier's principle. a** Comparison of self-healing mechanisms. (i) Conventional self-healing materials such as polymers, glasses, and metals. (ii) Self-healing (photo)catalysts such as photoelectrodes and air-reactive catalysts. (iii) Our newly developed self-healing system. In the system, self-healing reactions on the material occur spontaneously without external inputs. **b** Schematic illustration of self-healing mechanism based on Le Chatelier's principle. When equilibrium is disturbed due to material damage (first step), the reaction system tends to shift to mitigate the disturbance (second step). Following the completion of the self-healing process, the reaction system returns to equilibrium (third step).

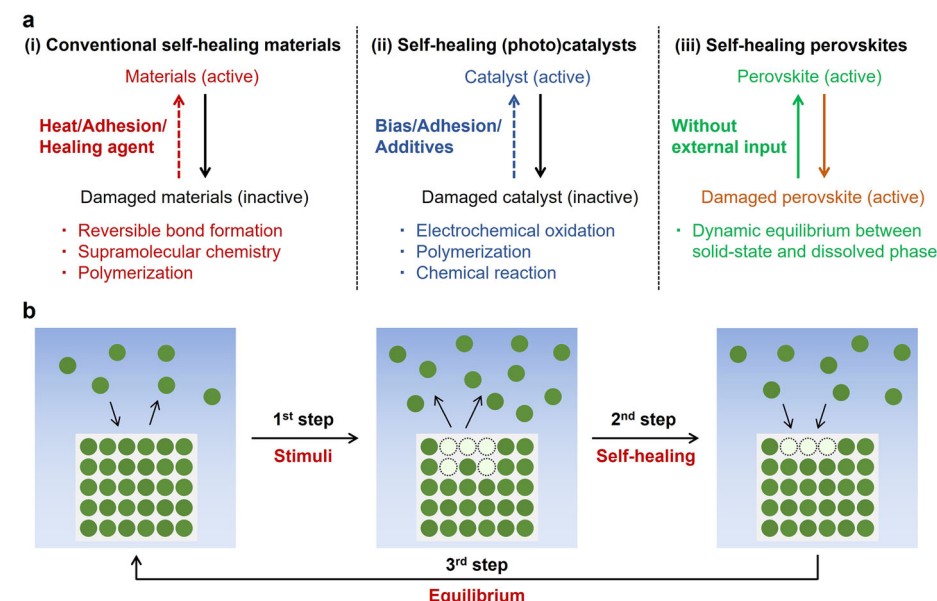

**Fig. 2 | Single-particle observations using microscopic techniques. a** Experimental setup for single-particle measurements. The experimental setup is based on a wide-field fluorescence microscope system. Both PL and transmitted light were captured using the same objective lens. **b–f** Optical images of mixed-halide MAPbBr$_x$I$_{3-x}$ perovskites with various $x$-values in aqueous solution. **g–k** Optical transmission images in aqueous solution. The inset labels indicate the time after the start of photoirradiation. Excitation was provided for the first 300 s, after which photoirradiation was stopped. A 405-nm CW laser (ca. 780 mW·cm$^{-2}$ at the sample surface) was used as the excitation source.

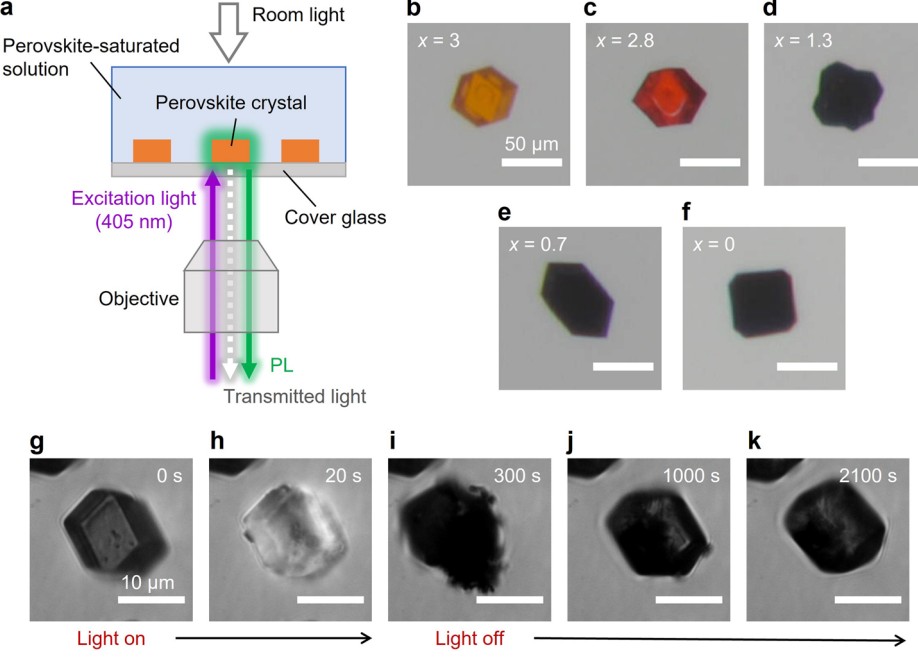

these conditions, they can act as photocatalysts for hydrogen generation under visible light irradiation[22,23]. We evaluated the self-healing behavior of perovskites in dynamic equilibrium at the single-particle level using a fluorescence microscope. We demonstrated that photodamaged crystals in the system possess self-healing abilities without requiring any external input. Interestingly, even in their damaged state, perovskites continue to produce hydrogen gas in aqueous solution, indicating that they can function as (photo)catalysts in both healed and damaged states. These findings suggest a possible strategy for the development of sustainable systems.

## Results and discussion
### Single-particle observations of crystal destruction and self-healing

Halide perovskites are unstable in the presence of humidity[24,25], oxygen[26,27], and heat[28,29]. However, when they are in a dynamic equilibrium state, the effects of these stimuli may be mitigated because the saturated solution acts as a passivating medium. Therefore, we employed single-particle fluorescence microscopy to directly monitor structural changes in individual perovskite crystals in response to light irradiation. This technique allows the evaluation of nanoscopic heterogeneities in chemical reactions at solid-liquid interfaces, which are typically obscured in ensemble-averaging bulk measurements[30–32].

Figure 2a depicts the experimental setup for single-particle observation using a home-built inverted fluorescence microscope system[33,34]. MAPbBr$_x$I$_{3-x}$ microcrystals (where MA$^+$ = CH$_3$NH$_3^+$) were spin-coated onto a cleaned cover glass to serve as seed crystals. A supersaturated perovskite solution was prepared by heating a suspension containing the target crystals at 80 °C, which was immediately dropped onto the substrate. As the temperature of the solution decreased, the perovskite crystals reprecipitated owing to the reduced solubility. Few studies have been conducted on the

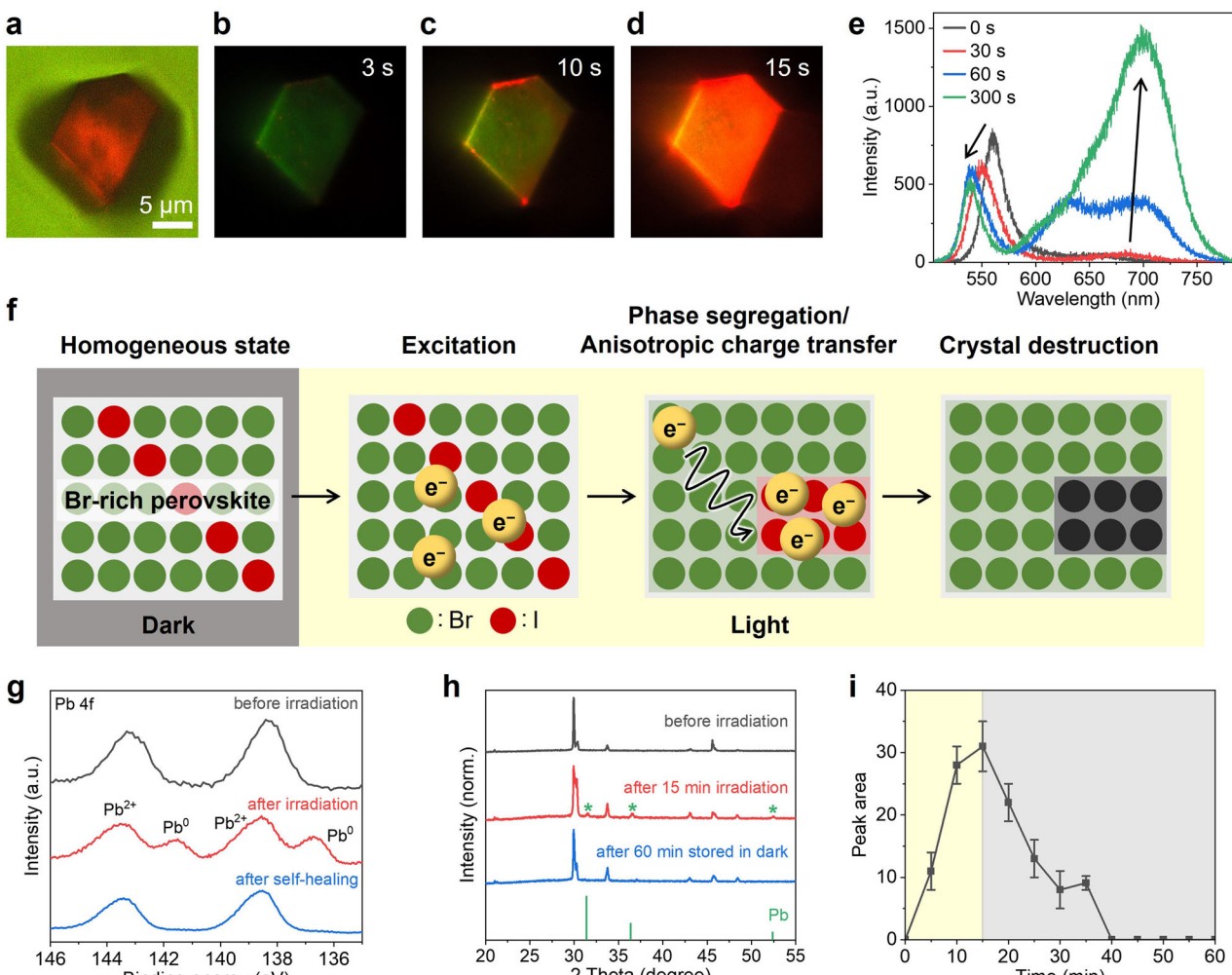

**Fig. 3 | Damaging and self-healing reactions of perovskites.** Optical transmission and PL images of MAPbBr$_{2.8}$I$_{0.2}$ in aqueous solution before (**a**) and during photo-irradiation (**b–d**). The inset labels show the time elapsed from the start of light irradiation. A 405-nm CW laser (ca. 1.21 W·cm$^{-2}$ at the sample surface) was used as the excitation source. **e** Spectral changes of the MAPbBr$_{2.8}$I$_{0.2}$ crystal in aqueous solution under irradiation. A 405-nm pulsed laser (ca. $8\times10^{-17}$ J·pulse$^{-1}$) was used as excitation source. **f** Schematic illustration of the crystal destruction mechanism induced by halide phase segregation. **g** Pb 4 f XPS spectra of MAPbBr$_{2.8}$I$_{0.2}$ before and after irradiation, and after self-healing. A 405-nm LED (ca. 350 mW·cm$^{-2}$) was used as excitation source. **h** XRD patterns of MAPbBr$_{2.8}$I$_{0.2}$ in aqueous solution before and after irradiation. A 405-nm LED (ca. 350 mW·cm$^{-2}$) was used as excitation source. Asterisks indicate characteristic peaks of metallic Pb (PDF card: 00-004-0686). **i** Temporal changes in the Pb$^0$ peak area under irradiation (yellow-shaded region) and after stopping irradiation (gray-shaded region). The error bars represent the standard deviation.

synthesis of perovskites in aqueous media[35–37]. Therefore, we investigated the correlation between the halide composition of the prepared solution and that of the resulting crystals. Interestingly, we found that iodide anions tended to be incorporated into the crystals more readily than bromide anions (Supplementary Fig. 1a). A better correlation was observed between the iodide/lead ratios of the prepared solution and those of crystals (Supplementary Fig. 1b). A discussion of this tendency is provided in Supplementary Note 1. After allowing the crystals to grow sufficiently, we initiated imaging of the crystal morphology and spectroscopic measurements (Supplementary Fig. 1c). Figure 2b–f illustrates optical images of the obtained MAPbBr$_x$I$_{3-x}$ crystals under dynamic equilibrium. The halide compositions were determined based on data from X-ray diffraction (XRD)[38] (Supplementary Fig. 1d). Photoluminescence (PL) imaging of individual perovskite crystals with various halide compositions revealed that the size of the MAPbBr$_3$ and MAPbI$_3$ crystals remained unchanged, or slightly decreased, or increased during 405-nm continuous wave (CW) laser irradiation (Supplementary Figs. 2 and 3). However, in the case of the mixed-halide MAPbBr$_{2.8}$I$_{0.2}$ crystals, the crystal morphology changed drastically when the excitation density was at least 780 mW·cm$^{-2}$ or higher

at the sample surface (Fig. 2g–i and Supplementary Movie 1), and we refer to this morphological change as crystal destruction. When the damaged crystals in the aqueous solution were left in the dark after crystal destruction, self-healing behavior was observed (Fig. 2j, k and Supplementary Movie 1). This finding indicates that the proposed self-healing mechanism based on dynamic equilibrium applies to perovskites.

## Mechanisms of crystal destruction and self-healing

Next, we investigated the mechanisms underlying the observed crystal destruction and self-healing behaviors. To determine why only the mixed-halide MAPbBr$_{2.8}$I$_{0.2}$ crystals were destroyed, we conducted color imaging of the perovskites under laser irradiation. Figure 3a, b–d displays the optical transmission image of a MAPbBr$_{2.8}$I$_{0.2}$ crystal before laser irradiation and the PL images of MAPbBr$_{2.8}$I$_{0.2}$ under 405-nm CW laser irradiation, respectively. Initially, the MAPbBr$_{2.8}$I$_{0.2}$ crystal exhibited a weak green emission. As the laser irradiation time increased, the edges of the crystal began to exhibit red emission. Finally, a bright red emission was observed from the entire crystal. Spectral measurements of a specific region (approximately 2 µm$^2$ in area) of the crystal revealed that MAPbBr$_{2.8}$I$_{0.2}$

initially exhibited PL at approximately 560 nm. As the irradiation time increased, however, this peak exhibited a slight blue shift and a broad emission band appeared at longer wavelengths. The intensity of the latter component increased and red-shifted, eventually reaching a maximum at approximately 700 nm (Fig. 3e and Supplementary Fig. 4). These characteristic PL changes were observed repeatedly (Supplementary Fig. 5) and were attributed to light-induced phase segregation[39]. To explain the origin of phase segregation, several models have been proposed[40], including those based on the miscibility gap due to thermodynamically unstable mixed-halide states, polaron-induced lattice strains that increase the mixing enthalpy of halide anions, and electric field-driven ion migration induced by the trapping of charge carriers at defects. In any cases, based on the band structures of perovskites, photogenerated charge carriers migrate from bromide-rich domains to iodide-rich domains with a narrower band gap, leading to the appearance of red-shifted PL (Supplementary Fig. 6). Similar PL features related to phase segregation have been reported elsewhere[41-43]; however, these studies did not involve conditions of aqueous dynamic equilibrium. We also conducted the same measurements on $MAPbBr_{0.3}I_{2.7}$ and $MAPbBr_{1.3}I_{1.7}$ (Supplementary Fig. 7). $MAPbBr_{1.3}I_{1.7}$ exhibited a similar red-shifted PL, followed by crystal destruction and self-healing reactions. In contrast, $MAPbBr_{0.3}I_{2.7}$ did not show such behavior and retained their original morphology without any damage. Furthermore, we examined whether phase segregation induces the destruction of crystals with extreme halide compositions. In this experiment, a supersaturated HI (HBr) solution was dropped onto a $MAPbBr_3$ ($MAPbI_3$)-coated cover glass. Hereafter, we refer to each resulting crystal as $MAPbI_3$ ($MAPbBr_3$) with a small amount of bromide (iodide) based on the PL peak at approximately 760 nm (540 nm) (Supplementary Figs. 8a and 9a). Upon exposure to light, the morphology of $MAPbI_3$ with a small amount of bromide remained unchanged (Supplementary Fig. 8b, c), whereas the $MAPbBr_3$ crystals with a small amount of iodide were destroyed in a manner similar to that of $MAPbBr_{2.8}I_{0.2}$ (Supplementary Fig. 9c, d). As illustrated in Supplementary Fig. 9b, the PL spectrum of $MAPbBr_3$ with a small amount of iodide not only exhibits dominant green PL at approximately 540 nm but also weak red PL at approximately 700 nm, indicating that light-induced phase segregation occurred even with an extremely small amount of iodide anions. These results indicate phase segregation induces the crystal destruction. Therefore, self-healing effect is not exclusive to the $MAPbBr_{2.8}I_{0.2}$, it can be applicable to other composition perovskites.

Based on the aforementioned results, we conclude that phase segregation is essential for crystal destruction (Fig. 3f). Prior to light irradiation, halide anions are uniformly distributed within the crystal. Upon photo-excitation of the mixed-halide perovskites, phase segregation mainly occurs in the regions near the surface, considering the limited penetration depth (~ 120 nm) of the excitation light[44]. Holes are trapped by chemical species such as halide anions in solution[23], leading to the accumulation of excess electrons in the localized iodide-rich regions. The accumulated electrons reduce the perovskites, leading to crystal destruction and morphological changes. Such significant crystal destruction was not observed in air, where no halide anions were present. The $MAPbI_3$ with a small amount of bromide did not decompose because the charge carriers remained within the iodide-rich regions that constitute most of the crystal (Supplementary Fig. 10). This proposed mechanism is supported by simultaneous imaging of the PL color and crystal morphology of $MAPbBr_{2.8}I_{0.2}$ crystals (Supplementary Fig. 11). Upon laser irradiation, $MAPbBr_{2.8}I_{0.2}$ emitted red PL, attributed to the iodide-rich domains formed by phase segregation. Once crystal destruction occurred, the regions near the damaged areas began to emit green PL, corresponding to bromide-rich domains, whereas red PL persisted in areas distant from the damage.

To assess the electronic states of $MAPbBr_{2.8}I_{0.2}$, we performed X-ray photoelectron spectroscopy (XPS). The Pb 4 f XPS spectra in Fig. 3g exhibit only $Pb^{2+}$ peaks before photoirradiation (black line), whereas $Pb^0$ peaks appeared after photoirradiation (red line). The formation and disappearance of metallic Pb were further confirmed by in-situ XRD measurements and scanning electron microscopy-energy dispersive X-ray

spectrometry (SEM-EDS) (Fig. 3h and Supplementary Fig. 12). In the XRD pattern, $Pb^0$ peaks were clearly observed, and no broad amorphous peaks appeared after photoirradiation. Therefore, we consider that most of the $Pb^0$ is in the crystalline phase. Figure 3i illustrates the temporal change in the Pb peak area. The perovskites were irradiated for 15 min, after which the light was turned off. With increasing photoirradiation time, the Pb peak area also increases. Photodegradation resulting from the generation of $Pb^0$ in mixed-halide perovskites via phase segregation has been reported previously[45,46], suggesting that our proposed mechanism is plausible. Such crystal destruction and self-healing reactions were further investigated using the diffuse reflectance spectra of $MAPbBr_{2.8}I_{0.2}$ (Supplementary Fig. 13). After photoirradiation, a broad band, possibly attributed to the absorption of photogenerated metallic Pb, was observed. This band nearly disappeared when the sample was kept in the dark.

Regarding the self-healing behavior in aqueous solution, we consider the dynamic equilibrium of perovskites as follows[21]:

$$MAPbBr_xI_{3-x}(s) \rightleftharpoons MA^+ + [PbBr_xI_{3-x}]^- \qquad (1)$$

In aqueous solution, various chemicals such as bromide anions, iodide anions, and phosphinic acids are present, with the latter being particularly important for reducing oxidative species such as triiodide anions. The standard redox potentials of the relevant reactions and the band structures of $MAPbX_3$ (X = Br, I) are summarized in Supplementary Fig. 14. The $Pb^0$ generated in the perovskite-saturated aqueous solution is oxidized to $Pb^{2+}$ according to the following reaction:

$$Pb^0 + 2H^+ \rightarrow Pb^{2+} + H_2 \qquad (2)$$

Protons are the dominant electron acceptors in aqueous solution because of the presence of phosphinic acid. The Gibbs free energy change ($\Delta G$) of Eq. (2) is determined by the following equation:

$$\Delta G = -nF\Delta E \qquad (3)$$

where $n$ is the number of electrons involved in the reaction, $F$ is Faraday's constant, and $\Delta E$ is the potential difference between the related half-reactions. The calculated $\Delta G$ is $-24.3$ kJ·mol$^{-1}$ [47], indicating that the oxidation of metallic $Pb^0$ to $Pb^{2+}$ proceeds spontaneously when $Pb^0$ is generated by the destruction of perovskites. Due to the high concentration of halide anions in the aqueous solution, lead complexes form[48].

$$Pb^{2+} + xBr^- + (3-x)I^- \rightarrow [PbBr_xI_{3-x}]^- \qquad (4)$$

Based on Le Chatelier's principle, the formation of $[PbBr_xI_{3-x}]^-$ causes a shift in the dynamic equilibrium, resulting in the production of $MAPbBr_xI_{3-x}(s)$, as expressed in Eq. (1). Consequently, the perovskite material self-heals. This self-healing behavior was also observed using in situ XRD measurements and diffuse-reflectance spectra. Upon stopping the light irradiation, the peak area associated with $Pb^0$ gradually decreased and eventually disappeared completely (Fig. 3i). After leaving damaged perovskites in the solution under dark condition, absorption of photons with lower than bandgap energy was drastically decreased, indicating that self-healing reactions are proceeded (Supplementary Fig. 13). Supplementary Fig. 15 shows PL spectra before photodamaging, after photodamaging, and after self-healing reactions. In this figure, the PL wavelength, which reflects the halide composition of perovskites, remained unchanged after the self-healing process. Hence, the photodamaging and self-healing processes do not affect the Br/I ratio of the crystal. Based on the formation of $MAPbBr_3$ from Pb powder in an aqueous HBr solution containing MABr as the organic cation source (Supplementary Fig. 16), the self-healing formation of mixed-halide perovskites is thermodynamically and kinetically feasible. This self-healing behavior, achieved without external stimuli, has significant implications for the development of novel catalysts that utilize a dynamic equilibrium reaction system.

**Fig. 4 | (Photo)catalytic hydrogen-production activity of perovskites under visible-light irradiation in aqueous solution. a** Optical images of the MAPbBr$_{2.8}$I$_{0.2}$ powder in saturated aqueous solution (left) before and (right) after 470-nm LED light irradiation (ca. 125 mW·cm$^{-2}$) for 24 h. **b** Total amount of hydrogen generated by perovskites, indicating the catalytic activity of MAPbBr$_3$ (black line and symbols), MAPbI$_3$ (red line and symbols), MAPbBr$_{1.3}$I$_{1.7}$ (purple line and symbols), MAPbBr$_{2.2}$I$_{0.8}$ (green line and symbols) and MAPbBr$_{2.8}$I$_{0.2}$ (blue line and symbols). The yellow-shaded region indicates the period of light irradiation, while the gray-shaded region indicates the period of darkness. **c** The hydrogen production of MAPbBr$_{2.8}$I$_{0.2}$ during intermittent irradiation. **d** Schematic illustration of degradation and self-healing reactions in aqueous solution under dynamic equilibrium. The self-healing reaction occurs spontaneously once the perovskites are damaged, and this cycle can continue repeatedly.

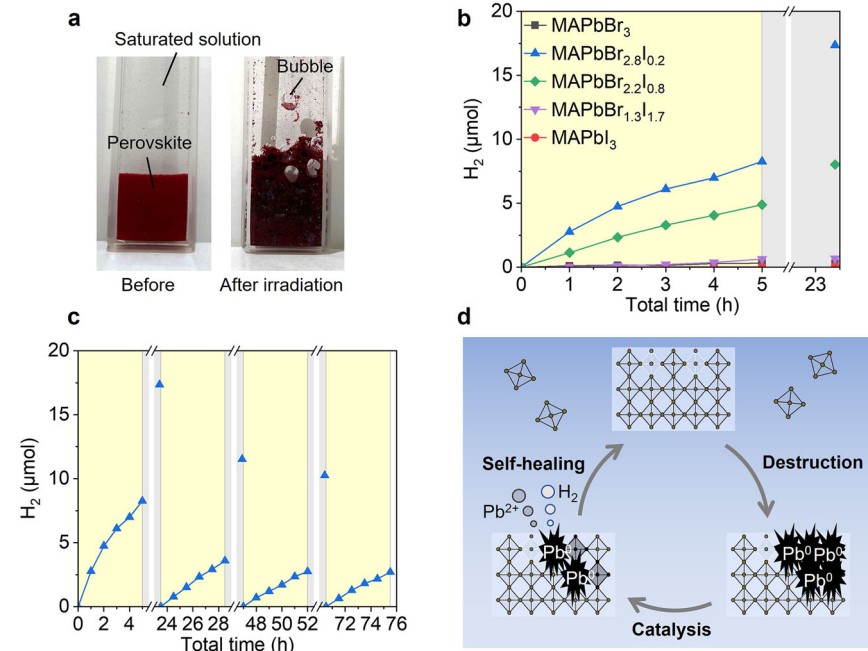

## Photocatalytic hydrogen evolution

Upon light exposure, the MAPbBr$_{2.8}$I$_{0.2}$ powder in a saturated aqueous solution exhibited a color change from reddish brown to black, due to the formation of metallic Pb (Fig. 4a and Supplementary Fig. 13)[49]. Moreover, numerous bubbles formed on the particles. As illustrated in Supplementary Fig. 14, the conduction band minimum of MAPbX$_3$ is more negative than the potential for proton reduction, enabling photocatalytic hydrogen generation as described in the following equations[23,36,50]:

$$MAPbBr_xI_{3-x} + h\nu \rightarrow MAPbBr_xI_{3-x}(h^+) + MAPbBr_xI_{3-x}(e^-) \quad (5)$$

$$MAPbBr_xI_{3-x}(2h^+) + 3I^- \rightarrow MAPbBr_xI_{3-x} + I_3^- \quad (6)$$

$$MAPbBr_xI_{3-x}(2h^+) + 2Br^- \rightarrow MAPbBr_xI_{3-x} + Br_2 \quad (7)$$

$$MAPbBrxI_{3-x}(2e^-) + 2H^+ \rightarrow MAPbBr_xI_{3-x} + H_2 \quad (8)$$

In addition, the photodamaging reaction occurred as follows:

$$MAPbBr_xI_{3-x}(2e^-) \rightarrow MA^+ + Pb^0 + xBr^- + (3-x)I^- \quad (9)$$

In the present dynamic equilibrium system, the self-healing reactions include hydrogen generation, as expressed in Eq. (2). Generally, photocatalysts are only active under photoirradiation. However, if our proposed mechanism holds true, perovskites can produce hydrogen not only when photoexcited but also under dark conditions. To test this hypothesis, we evaluated the photocatalytic hydrogen production activity using gas chromatography. Figure 4b compares the amounts of hydrogen gas produced during 5 h of visible light-driven HX splitting and 18.5 h of storage in the dark. In these experiments, the perovskites were able to produce hydrogen for 5 h, indicating that they can stably exist in the solution without complete crystal destruction for a longer duration compared to microscopic measurements. This is likely due to the difference in excitation power densities at the sample surface under each condition: ca. 780 mW·cm$^{-2}$ for microscopic measurements (Fig. 2) and ca. 125 mW·cm$^{-2}$ for photocatalytic measurements (Fig. 4). The photocatalytic activity of mixed-halide perovskites is significantly higher than those of the single-halide perovskites, MAPbBr$_3$

and MAPbI$_3$. One possible reason for the higher photocatalytic activity of the mixed-halide perovskites is light-induced phase segregation. During this process, iodide-rich domains are formed near the surface, enabling efficient charge transport to these domains and facilitating photocatalytic reactions[36]. This behavior explains how the original (undamaged) perovskites are effective photocatalysts. Hydrogen generation from the pre-irradiated samples held in the dark suggests that the damaged perovskites remained active during the self-healing reactions. As shown in Supplementary Fig. 17, the hydrogen generation rates under light are gradually decreased as light irradiation time increased, indicating that photocatalytic hydrogen evolution and self-healing-induced hydrogen evolution (Eq. (2)) occurred simultaneously. Another photocatalytic activity test supports our hypothesis. After initiating the photocatalytic reaction, we intermittently stopped the light irradiation. As illustrated in Supplementary Fig. 18, the system continued to exhibit stable hydrogen generation even in the dark. In our tests, the amount of Pb$^{2+}$ in the MAPbBr$_{2.8}$I$_{0.2}$ crystals was approximately 20.5 μmol. However, the perovskites are not completely reduced to Pb$^0$, as evidenced by their continued PL (Supplementary Fig. 11). This suggests that they retain part of their perovskite structure even after crystal destruction. Therefore, the actual amount of photogenerated Pb$^0$ should be less than 20 μmol. Nonetheless, the total amount of hydrogen produced under dark conditions reached approximately 24.5 μmol (9.08, 7.98, and 7.51 for the first, second, and third cycles, respectively, Fig. 4c), again highlighting the self-healing capability of MAPbBr$_{2.8}$I$_{0.2}$ as a photocatalyst. The estimation of the potential lifetime of the photocatalytic reaction system, based on the amount of chemicals in the solution, is discussed in Supplementary Note 2. The hydrogen production activity of mixed-halide perovskites tends to decrease as the ratio of iodide anions in the crystals increases (Fig. 4b). This may be because the increased iodide content leads to a larger area of phase-segregated iodide-rich regions. Consequently, less efficient electron transfer occurs compared to MAPbBr$_{2.8}$I$_{0.2}$, resulting in lower hydrogen production activity and reduced susceptibility to destruction. This explanation is supported by microscopic observations (Supplementary Figs. 7–9).

However, with an increasing number of cycles, their photocatalytic activity gradually decreased. This could be attributed to insufficient self-healing processes. Supplementary Fig. 19 shows SEM-EDS images of MAPbBr$_{2.8}$I$_{0.2}$ prepared under various conditions. Before irradiation, each component—Br, I, and Pb—was homogeneously distributed across the entire crystals. After photoirradiation, I-rich and Pb-rich regions,

corresponding to phase-segregated domains and photoreduced metallic Pb, respectively, were observed. Even after being left in their saturated solution, these I-rich and Pb-rich regions remained, suggesting that self-healing reactions were not completed within the experimental timescale. Furthermore, the solution may become supersaturated during irradiation due to the decomposition of crystals and the dissolution of their components.

Figure 4d summarizes the overall reaction scheme. Initially, perovskites stably exist in an aqueous solution by achieving dynamic equilibrium at the solid liquid interfaces. Upon photoexcitation, the accumulated electrons reduce $Pb^{2+}$ in the perovskites to $Pb^0$, resulting in significant morphological changes (Fig. 2i). After stopping light irradiation, self-healing reactions occur along with hydrogen generation. The reconstructed (self-healed) perovskites again function as photocatalysts for continued use.

Supplementary Fig. 20 illustrates the cycle of deciduous trees[51]. When temperatures cool, dormancy is induced, leading to the transfer of nitrogen (N), an energy source for plants, from leaves to stems. Subsequently, the leaves are shed. The stored nitrogen is then used during bud break in the next cycle to produce new growth. The self-healing mechanism of perovskites demonstrated in this study is analogous to this process. In the damaged state, akin to the dormant state in plants, perovskites store energy in the form of charges in the metallic $Pb^0$ regions. The self-healing reaction, which is comparable to bud break and leaf growth in plants, is driven by the utilization of both stored chemical energy and thermal energy. The similarity between the processes in perovskites and plants suggests that biomimicry is a powerful strategy for achieving efficient energy utilization.

## Conclusions

In summary, we developed novel self-healing systems incorporating mixed-halide perovskites, based on Le Chatelier's principle. Unlike other self-healing materials or systems, the proposed mechanism does not require external input to induce a self-healing reaction under dynamic equilibrium conditions. We further demonstrated that this self-healing system applies to photocatalytic hydrogen generation. Even if the self-healing perovskites are damaged by prolonged light exposure, they can stably produce hydrogen in the dark. This self-healing capability is particularly beneficial for the practical application of photocatalytic systems, as it accounts for the day-night cycle on our planet. During the day, the materials function as photocatalysts, and at night, they heal. The dynamic transition between the solid state and the dissolved state is fundamental to our self-healing mechanism, thereby making it applicable to a wide range of materials, including those based on non-covalent bonds (e.g., metal-organic frameworks and organic crystals). We believe that our proposed mechanism could serve as a design principle for innovative and sustainable self-healing materials and systems for photocatalysis and other applications.

## Methods
### Chemicals

Hydriodic acid (HI) (Sigma-Aldrich; 57 wt% in water, 99.99% trace metal basis), lead (Pb) (Sigma-Aldrich; 99.5%), methylamine hydroiodide (MAI) (TCI; low water content, >99.0%), lead(II) iodide ($PbI_2$) (TCI; 99.99%, trace metal basis, >98.0%), methylamine hydrobromide (MABr) (TCI; low water content, >98.0%), lead(II) bromide ($PbBr_2$) (TCI; >98.0%), γ-butyrolactone (TCI; >99.0%), phosphinic acid solution (FUJIFILM Wako Chemicals; 50 wt% in water), hydrobromic acid (HBr) (FUJIFILM Wako Chemicals; 47.0–49.0 wt% in water), toluene (FUJIFILM Wako Chemicals; super dehydrated, >99.5%), N,N-dimethylformamide (DMF) (FUJIFILM Wako Chemicals; super dehydrated, >99.5%), 2-propanol (FUJIFILM Wako Chemicals; super dehydrated, >99.7%), were used without further purification.

### Synthesis of MAPbX₃ microcrystals by using an aqueous solution and preparation of MAPbX₃-saturated HX solution (X = Br, I)

Equimolar amounts of MAX and $PbX_2$ were added to the same amount of $HX/H_3PO_2$ solution (HX : $H_3PO_2$ = 4 : 1, v/v), respectively. These solutions were sonicated, and then MAX solution was added to the $PbX_2$ solution to

initiate the crystallization of $MAPbX_3$. The mixed solution was heated in an oil bath at 100 °C for 1 h. After heating, the solution was allowed to cool to room temperature. The supernatant solution was collected and stored as $MAPbX_3$-saturated HX solution for further experiments. The obtained crystals were washed several times with toluene and then dried in a vacuum dryer at 45 °C.

### Synthesis of MAPbBrₓI₃₋ₓ microcrystals by using an aqueous solution and preparation of MAPbBrₓI₃₋ₓ-saturated HBr/HI solution

$MAPbBr_3$ synthesized via aqueous media was used as seed crystals. A certain amount of $MAPbBr_3$-saturated HBr solution and $MAPbI_3$-saturated HI solution were added to $MAPbBr_3$ crystals to mix halide anions. The mixed solution was heated in an oil bath at 100 °C for 30 min. After heating, the solution was allowed to cool to room temperature. The supernatant solution was used as $MAPbBr_xI_{3-x}$-saturated HBr/HI solution for further experiments. The obtained crystals were washed several times with toluene and then dried in a vacuum dryer at 45 °C.

### Synthesis of MAPbBr₃ microcrystals by using an organic solvent

81.8 mmol of MABr and $PbBr_2$ were added to 2 mL of DMF and the solution was sonicated to completely dissolve the precursors. 4 mL of toluene was slowly added dropwise to the precursor solution. The obtained crystals were separated by centrifugation and washed several times with toluene.

### Synthesis of MAPbI₃ microcrystals by using an organic solvent

0.25 mmol of MAI and $PbI_2$ were each added to 1 mL of 2-propanol, and both solutions were sonicated. After sonication, the $PbI_2$ solution was added dropwise to the MAI solution. The obtained crystals were separated by centrifugation and washed several times with toluene.

### Synthesis of MAPbBrₓI₃₋ₓ microcrystals by using an organic solvent

To prepare the bromide-precursor solution, 81.8 mmol of MABr and $PbBr_2$ were added to 8 mL of γ-butyrolactone. Similarly, 81.8 mmol of MAI and $PbI_2$ were added to 3 mL of γ-butyrolactone to prepare the iodide-precursor solution. Each precursor solution was sonicated to completely dissolve the chemicals. A certain amount of bromide-precursor and iodide-precursor solutions were then mixed, ensuring the total volume of the mixed solution was 2 mL. Subsequently, 4 mL of toluene was slowly added dropwise to the precursor solution. The obtained crystals were separated by centrifugation and washed several times with toluene.

### Characterizations

X-ray diffraction (XRD) patterns were measured on X-ray diffractometer (Rigaku, MiniFlex) with Cu Kα radiation. The lattice parameters were calculated by Rietveld analysis using SmartLab Studio II Powder XRD software (Rigaku), and the halide compositions were determined by the lattice parameters. X-ray photoelectron spectroscopy (XPS) measurements were conducted using a photoelectron spectrometer (ULVAC-PHI, PHI X-tool) to analyze the electronic states of the perovskites. Steady-state diffuse reflectance spectra were measured on UV-visible-NIR spectrophotometer (JASCO, V-770). A zoom stereomicroscope (Nikon, SMZ800N) equipped with a microscope camera (Nikon, Digital Sight 1000) was used to image the samples. Scanning electron microscopy (SEM) combined with energy dispersive X-ray spectroscopy (EDS) (JEOL, JCM-7000) was used to analyze the structure and composition of the materials.

### Sample preparation for single-particle photoluminescence (PL) experiments

Cover glasses were purchased from Matsunami-glass, Ltd. and cleaned by sonication in a 20% alkaline detergent solution (AS ONE Corporation, Cleanace) for several hours. The glasses were then washed ten times with distilled and ultrapure water (Milli-Q). Perovskite microcrystals synthesized

by using an organic solvent were spin-coated onto the cleaned cover glass as seed crystals. A perovskite-supersaturated HX solution was then dropped onto the perovskite-coated cover glasses to induce crystal growth as the solution cooled. Photoluminescence (PL) measurements were carried out on the grown crystals.

## Single-particle PL measurements

Single-particle PL measurements were conducted using a home-built inverted fluorescence microscope (Nikon, Ti-E) system. During the measurements, we covered the sample holder to prevent solution volatilization and maintained the room temperature at $23 \pm 1$ °C to preserve dynamic equilibrium. A 405-nm continuous wave (CW) laser (Coherent, OBIS 405LX) or picosecond pulsed laser (Advanced Laser Diode Systems, PiL040X) was used as the excitation source which was focused through an oil-immersion objective lens (Nikon, CFI Plan Apo λ 100× Oil; NA 1.45). The emission from the samples was collected using the same objective lens. A dichroic mirror (Semrock, Di02-R405) and a longpass filter (Semrock, BLP01-458R) were used to block scattered excitation light. Additional fluorescence filters were employed to obtain the appropriate images. An electron-multiplying CCD (EMCCD) camera (Photometrics, Evolve 512) or a color sCMOS camera (Tucsen Photonics, Dhyana 400DC) was used to capture the transmission and PL images. PL spectra were measured using an imaging spectrograph (SOL instruments, MS 3504i) equipped with a CCD camera (Andor, DU416A-LDC-DD). Data were analyzed using ImageJ (https://rsb.info.nih.gov/ij/) and OriginPro 2023 (OriginLab).

## H_2 production activity tests

A gas chromatograph (Shimadzu, GC-8A) equipped with an MS-5A column and a thermal conductivity detector (TCD) was used to measure the amount of produced $H_2$ gas. 10 mg of $MAPbBr_xI_{3-x}$ was dispersed in 5 mL of $MAPbBr_xI_{3-x}$-saturated HBr/HI solution. The suspensions were purged with Ar gas for 5 min to remove dissolved oxygen, then sealed with a rubber septum. For photocatalytic $H_2$ production, the suspension was exposed to visible light (ca. 125 mW·cm$^{-2}$) from a 470 nm LED (Thorlabs, M470L3; nominal wavelength = 470 nm, bandwidth (FWHM) = 25 nm) through a 420 nm sharp cut filter (OptoSigma, SCF-50S-42L). The suspension was vigorously stirred during the experiment at room temperature. The amounts of photogenerated $H_2$ gas were measured using a gas chromatograph. During the measurements, we sealed the reactor to prevent solution volatilization and maintained the room temperature at $23 \pm 1$ °C to preserve dynamic equilibrium.

## Data availability

All data that support the conclusions of this study are available in the paper and the Supplementary Information or from the corresponding author on reasonable request. Source data presented in main manuscript are provided in Supplementary Data 1. The result of microscopic imaging during photodamaging and self-healing processes is provided in Supplementary Movie 1.

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

## Acknowledgements

This work was partly supported by the Iwatani Naoji Foundation's Research Grant, JSPS KAKENHI (Grant Nos. JP23KK0097 and JP24K21753) and JST SPRING (Grant No. JPMJSP2148). We would like to thank Editage (www.editage.jp) for English language editing.

## Author contributions

A.T. synthesized the materials. A.T. and Y.K. characterized the materials and A.T. and T.T. performed single-particle experiments. A.T. wrote the draft of the manuscript. All authors discussed the results and contributed to the final manuscript. T.T. conceived and supervised the project.

## Competing interests

The authors declare no competing interests.
