## [Transparent Peer Review file · Communications Chemistry]

Unassisted self-healing photocatalysts based on Le Chatelier's principle

Corresponding Author: Professor Takashi Tachikawa

Version 0:

Reviewer comments:

Reviewer #1

(Remarks to the Author)

In this manuscript, the authors described an unassisted self-healing mechanism based on the dynamic equilibrium between the solid-state and dissolved materials and employed organic-inorganic perovskite photocatalysts to validate this strategy. Overall this study is interesting and of some practical importance to extending the lifetime of a photocatalyst. It can be considered for publication after minor revisions.

1. In Fig. 3g, I suggest to test one more sample for XPS analysis: the sample after 60 min stored in dark, the same sample indicated as the blue line in Fig. 3h.

2. In Fig. 3h, what does the peak at ca. 33-34° indicate? This peak seems to increase slightly after 15 min irradiation and to remain after 60 min stored in dark. The increment is more obvious when comparing this peak with the one at ca. 45°. But the authors did not discuss this peak.

Minor issues:

L 199, "Pb⁰" here "0" should be superscript.

L 226, "kJ mol⁻¹" should be "kJ·mol⁻¹".

L231 and 232, "MAPbBr_{3-x}" here "x" should be italic typeface.

Reviewer #2

(Remarks to the Author)

The authors report a very interesting and detailed investigation of the mixed-halide hybrid perovskite nanocrystals and reveal their light-assisted destruction and reassembly mechanisms. I find the study to be executed and presented on a very high level, with thoughtful and elaborate experiments conducted to validate their hypotheses. The results look very convincing; however I have several comments to the presentation of this work, which I would like the authors to consider and address in their revision.

1. My main concern is related to the way the authors refer to their material as a photocatalyst and the process of H₂ production as photocatalytic. The field of photocatalysis is a very established one, and the community (especially in Japan) would very much agree on what is photocatalytic HER of a semiconductor. It is first of all a catalytic process. From the reported results, I, however, see that both the processes of H₂ production (during perovskite destruction and its self-healing) are rather stoichiometric chemical reactions that result in H₂. They are clearly triggered by light, but I don't see proof that they are truly catalytic. The authors agree with the fact that the second process (healing) is a chemical reaction in which Pb⁰ species (that acted as an electron reservoir) are able to release the electrons resulting in HER (eq 2.). My proposal is that the first process (destruction of the perovskite) should also be classified as a light-activated/driven process, not as a (photo)catalytic one. Based on Figure 4c, the H₂ production under light is constant and stable, but the overall amount of H₂ produced in these illumination cycles is never exceeding that of the self-healing (dark) H₂ generation. This means that the amount of H₂ formed never reaches the stoichiometric number of species in the perovskite (Pb/halides etc). So, the catalytic nature cannot be confirmed as of now. The light-driven H₂ generation (first cycle) can be simply associated with the degradation mechanism, which triggers H₂. I invite the authors to consider this. I would just propose not to refer to the process as (photo)catalytic, which, in my opinion, will not affect the story and messages of this work.

2. I also find it unnecessary to showcase the use of the perovskite with Pt/TiO₂. One of the main reasons for the enhanced HER when using Pt/TiO₂ can be the presence of Pt itself. So, I find the conclusion that the presence of Pt/TiO₂ can enhance the self-healing properties of the perovskite to be superficial. This section raises more questions that it gives answers or sheds light. I would thus propose to remove it. Page 14 line 292-301

3. One question regarding the “photocatalytic” test setup: the authors used 5 h of illumination, during which H₂ production rate was almost constant. But previously you show that the destruction of the crystal is very quick (of course under different irradiation conditions but nevertheless). What do you think is happening during the 5 h of illumination? Is the destruction complete soon after the illumination starts (within seconds or minutes), but a certain crystalline core remains, or do you think the destruction proceeds slowly over the period of 5 hours? If latter – when is the final state of the destruction reached? Also, for someone working with photocatalytic setups, the natural question is the difference in light sources used and light intensities applied between your catalytic studies and microscopy investigations. It would be important to comment on this in the main text or experimental section.

4. I think a few small clarifications/statements would be required to be added to the manuscript text for these points:

- Page 6, line 115: It is somewhat natural to assume that the healing can happen in such supersaturated solutions where precursor molecules are always available for the crystal to re-grow/heal. The authors don't really comment on this but maybe the seed (core) always stays intact, while the outer shell is mostly destroyed during the illumination. If this is the case and the crystalline core always stays intact, it will naturally promote that the crystal grows back (to its original form). I would also assume that if stirring is applied and there is time for the precursor species after the destruction to diffuse away, such re-growth would not be observed. Similarly, if the authors would work with smaller crystals, would one expect their full destruction and that no such healing would take place (since no core would be preserved).

- Can it also be that the crystal does not dissolve completely (you see a dark spot) but rather amorphise?

- In Figure S4 one can also see an intermediate emissive state (120-180 sec) which the authors don't comment on

5. Generally, I really appreciate that the authors to report laser intensities, but could you pls comment on the importance of these / their control? Have you observed some of the effects only at specific intensity ranges? If yes, please add this discussion and your interpretation. Same goes to the relevance of the effects seen in the microscope vs under photocatalytic conditions.

6. Maybe some clarification would be needed here: in the microscopy setup light is used to probe PL, but also light is used to trigger destruction. Are these effects / actions separated? How do the authors ensure that data acquisition takes place under “steady-state” conditions? It is not clear from the text and discussions

Reviewer #3

(Remarks to the Author)

In this study, the authors explore the photoinduced destruction and healing, over multiple cycles, of a perovskite material using MAPbBr_{2.8}I_{0.3} as a model system. The authors present a well-informed hypothesis based on their observations supported by several results of their characterization techniques, specifically, single particle spectroscopy. The authors propose the Le-chatelier's principle as the driving force towards this self healing ability of the material. Finally, the authors demonstrate the use of such materials towards H₂ production from HX (X = Br, I), both in the presence and absence of visible light – thereby suggesting a photosystem working based on the day-night cycle on earth. While the authors also attempt to rule out other factors using a set of reference experiments, I believe there are a few important questions that need to be addressed that can further improve the clarity of their message overall.

Major comments:

Questions regarding the self-healing mechanism and experiments

Q1: While the authors propose the Le-chatelier's principle as the reason driving the self healing process, the perovskite MAPbBr_{2.8}I_{0.2} only demonstrates the self healing property in the absence of light (as mentioned in Lines 113 and 114), as opposed to a simultaneous healing process. Can the authors comment why such a behaviour is observed? Would they not expect a more immediate self-healing process due to the changes induced, according to the Le-chatelier's principle?

Q2: Does the overall Br / I ratio remain the same after multiple cycles of healing, or is that changed depending on the surrounding solution that only contains one of the anions available?

Q3: The authors suggest the destruction to be surface sensitive, however, ESI Figure 11 shows the phase segregation in the whole of the crystal? why is that? Is the crystal very small compared to light penetration depth?

Questions regarding photocatalysis

Q4: While there is plenty of text in the manuscript on the proposed H₂ production mechanism, I would recommend the authors to present a few reaction equations showing the oxidation and reduction reactions involved giving the reader a clear idea of the role of the photogenerated electrons and holes in the process.

Q5: While the authors show the band diagrams of the pure Br₃ and I₃ perovskites to be capable of the hydrogen evolution reaction (Figure S14), the observed CB values are far below (more negative) vs the mixed halide composites. Why in this case do Br₃ and I₃ compounds underperform in HER?

Q6: The photocatalytic performance of the best performing perovskite MAPbBr_{2.8}I_{0.2} seems to be around 2.5 μmol of H₂ produced in the first hour (Figure 4b). However, in the reference photocatalytic experiment results shown in Supplementary

Figure 16, the material seems to only produce $\sim 1.2 \mu\text{mol}$ of H_2 in the first hour. Could the authors comment on such a fluctuation observed in the initial photocatalytic performance?

Q7: while the 1.3/1.7 shows red shift in PL the 0.3/2.7 does not. Could the authors give any comment as to why that happens?

Minor comments:

Q8: It seems that in ESI Fig 8 and 9 the intensity of the peak increases over illumination time for the I-perovskite and not really for the Br perovskite. Neither does the intensity of Br perovskite go down upon destruction – why is that?

Q9: Figure S9: The sliced PL at 241 s shows the segregation but, in the time-plot, the intensity of 700 nm band at 300 s seems to have gone back to 0. Could the authors add the slice at 300 s as well, and comment on this dataset (also relevant to Line 161-164)

Q10: In the discussion relating to phase segregation as the reason behind crystal destruction, in Line 186, they say that the holes are trapped by the halide ions in the solution and therefore the electrons (excess) reduce the perovskite and destroy it. Would this also occur if the perovskite would not be immersed in halides saturated solution (in this case there is no halide ions to trap holes)? This would mean that both healing AND DESTRUCTION would require this specific conditions. Please clarify

Q11: Line 204, clarify which Pb oxidation state is mentioned.

Q12: Fig 4b missing x axis legend

Version 1:

Reviewer comments:

Reviewer #1

(Remarks to the Author)

The authors have addressed all my concerns and hence this manuscript can be considered for publication.

Reviewer #2

(Remarks to the Author)

I think the authors have carefully considered my comments and made respected changes in the MS and the ESI. I recommend this work for publication.

Reviewer #3

(Remarks to the Author)

I am satisfied with the response from the authors and the corresponding changes in the manuscript. I would recommend for publication without any further changes.

Responses to the Comments Given by Reviewer #1:

In this manuscript, the authors described an unassisted self-healing mechanism based on the dynamic equilibrium between the solid-state and dissolved materials and employed organic-inorganic perovskite photocatalysts to validate this strategy. Overall this study is interesting and of some practical importance to extending the lifetime of a photocatalyst. It can be considered for publication after minor revisions.

Response) We appreciate the reviewer for your high evaluation of our work. We have done our best to revise our manuscript according to your suggestion as follows.

1. In Fig. 3g, I suggest to test one more sample for XPS analysis: the sample after 60 min stored in dark, the same sample indicated as the blue line in Fig. 3h.

Response) Thank you very much for your valuable comment. We analyzed the sample after self-healing reactions using XPS and have added the result to Fig. 3g.

2. In Fig. 3h, what does the peak at ca. 33-34° indicate? This peak seems to increase slightly after 15 min irradiation and to remain after 60 min stored in dark. The increment is more obvious when comparing this peak with the one at ca. 45°. But the authors did not discuss this peak.

Response) Thank you very much for your valuable comment. There are several possible explanations for the increase in peak intensity. One possibility is a change in crystal facet alignment due to photo-damaging reactions. Another possible reason is that crystal damage mainly occurs in iodide-rich domains. The peak at 33-34° can be assigned to the (210) plane of bromide perovskites. As a result, the peak corresponding to the undamaged bromide perovskites appears to have increased. Of course, the changes in peak intensity are an important point and should be discussed. However, it is difficult to determine the exact reason based on our experimental results because we removed the sample from the XRD instrument after the “before irradiation” measurement (shown in black in Fig. 3h) to irradiate it with light to induce photo-damaging reactions.

Minor issues:

L 199, “Pb0” here “0” should be superscript.

L 226, “kJ mol⁻¹” should be “kJ·mol⁻¹”.

L231 and 232, “MAPbBrxI3-x” here “x” should be italic typeface.

Response) Thank you very much for your valuable comment. We have made the corrections.

Responses to the Comments Given by Reviewer #2:

The authors report a very interesting and detailed investigation of the mixed-halide hybrid perovskite nanocrystals and reveal their light-assisted destruction and reassembly mechanisms. I find the study to be executed and presented on a very high level, with thoughtful and elaborate experiments conducted to validate their hypotheses. The results look very convincing; however I have several comments to the presentation of this work, which I would like the authors to consider and address in their revision.

Response) Thank you very much for your recommendation following the revisions and your thoughtful consideration of our work. Our paper was significantly improved by your helpful advice. We have done our best to revise our manuscript according to your suggestion as follows.

1. My main concern is related to the way the authors refer to their material as a photocatalyst and the process of H₂ production as photocatalytic. The field of photocatalysis is a very established one, and the community (especially in Japan) would very much agree on what is photocatalytic HER of a semiconductor. It is first of all a catalytic process. From the reported results, I, however, see that both the processes of H₂ production (during perovskite destruction and its self-healing) are rather stoichiometric chemical reactions that result in H₂. They are clearly triggered by light, but I don't see proof that they are truly catalytic. The authors agree with the fact that the second process (healing) is a chemical reaction in which Pb⁰ species (that acted as an electron reservoir) are able to release the electrons resulting in HER (eq 2.). My proposal is that the first process (destruction of the perovskite) should also be classified as a light-activated/driven process, not as a (photo)catalytic one. Based on Figure 4c, the H₂ production under light is constant and stable, but the overall amount of H₂ produced in these illumination cycles is never exceeding that of the self-healing (dark) H₂ generation. This means that the amount of H₂ formed never reaches the stoichiometric number of species in the perovskite (Pb/halides etc). So, the catalytic nature cannot be confirmed as of now. The light-driven H₂ generation (first cycle) can be simply associated with the degradation mechanism, which triggers H₂. I invite the authors to consider this. I would just propose not to refer to the process as (photo)catalytic, which, in my opinion, will not affect the story and messages of this work.

Response) Thank you very much for your valuable comment. If perovskites do not catalyze HER themselves, the generated hydrogen could be associated with oxidation of Pb^0 . In this case, the hydrogen production rate under light irradiation should gradually increase due to the accumulation of Pb^0 in the sample. However, our results suggest that hydrogen evolution activity under light tends to gradually decrease, indicating that the perovskites in our study can promote photocatalytic HER (Supplementary Fig. 17). At the same time, it is also possible that HER and perovskite-damaging reactions occur simultaneously. Based on these considerations, we have revised the manuscript in lines 286-288 and added Supplementary Fig. 17 to summarize the hydrogen production activity.

2. I also find it unnecessary to showcase the use of the perovskite with Pt/TiO₂. One of the main reasons for the enhanced HER when using Pt/TiO₂ can be the presence of Pt itself. So, I find the conclusion that the presence of Pt/TiO₂ can enhance the self-healing properties of the perovskite to be superficial. This section raises more questions than it gives answers or sheds light. I would thus propose to remove it. Page 14 line 292-301

Response) Thank you very much for your valuable comment. As you suggested, Pt/TiO₂ can work as HER sites, while the perovskites act as photosensitizers. According to your suggestion, we have removed Supplementary Fig. 18 and related descriptions.

3. One question regarding the “photocatalytic” test setup: the authors used 5 h of illumination, during which H₂ production rate was almost constant. But previously you show that the destruction of the crystal is very quick (of course under different irradiation conditions but nevertheless). What do you think is happening during the 5 h of illumination? Is the destruction complete soon after the illumination starts (within seconds or minutes), but a certain crystalline core remains, or do you think the destruction proceeds slowly over the period of 5 hours? If latter – when is the final state of the destruction reached? Also, for someone working with photocatalytic setups, the natural question is the difference in light sources used and light intensities applied between your catalytic studies and microscopy investigations. It would be important to comment on this in the main text or experimental section.

Response) We apologize for the insufficient description. The excitation conditions are quite different between microscopic (single-particle) and photocatalytic (ensemble-averaging) measurements. In microscopic measurements, we used an objective lens to focus the laser spot on a single perovskite crystal. In contrast, in photocatalytic measurements, the excitation light

was irradiated over a much larger area in the reactor compared to microscopic experiments. To clarify this point, we have added information about the excitation power density for each experiment in lines 273-278 and in the figure captions. Furthermore, the sample was stirred during photocatalytic activity tests, whereas in microscopic measurements, it was not. This induces significant charge accumulation in a single perovskite crystal under the microscope.

4. I think a few small clarifications/statements would be required to be added to the manuscript text for these points:

- Page 6, line 115: It is somewhat natural to assume that the healing can happen in such supersaturated solutions where precursor molecules are always available for the crystal to re-grow/heal. The authors don't really comment on this but maybe the seed (core) always stays intact, while the outer shell is mostly destroyed during the illumination. If this is the case and the crystalline core always stays intact, it will naturally promote that the crystal grows back (to its original form). I would also assume that if stirring is applied and there is time for the precursor species after the destruction to diffuse away, such re-growth would not be observed. Similarly, if the authors would work with smaller crystals, would one expect their full destruction and that no such healing would take place (since no core would be preserved).

Response) Thank you very much for your valuable comment. As you mentioned, the seed (core) plays a crucial role in initiating and facilitating self-healing reactions. If a perovskite crystal is excited with very high intensity, resulting in complete damage, it dissolves into the solution. Additionally, based on our experience, stirring accelerates self-healing reactions rather than inhibiting them. Therefore, we assume that several factors are closely related to self-healing reactions, specifically ensuring that the seed (core) remains intact and maintaining good contact between the seed (core) and the precursor solution. We have added a comment about this in the caption of Supplementary Fig. 16.

- Can it also be that the crystal does not dissolve completely (you see a dark spot) but rather amorphise?

Response) Thank you very much for your valuable comment. As shown in Fig. 3h (red line), Pb^0 peaks were clearly observed, and no broad amorphous peaks appeared after photoirradiation. Therefore, we consider that most of the Pb^0 to be in the crystalline phase. We have added a comment on this point in lines 205-207.

- In Figure S4 one can also see an intermediate emissive state (120-180 sec) which the authors don't comment on

Response) Thank you very much for your valuable comment. Light-induced halide phase segregation progresses as the light irradiation time increases. The emission wavelength of perovskite reflects their halide composition. Therefore, such an intermediate emissive state can be attributed to transient, partially phase-segregated perovskites. We have added a comment on this point in the caption of Supplementary Fig. 4.

5. Generally, I really appreciate that the authors to report laser intensities, but could you pls comment on the importance of these / their control? Have you observed some of the effects only at specific intensity ranges? If yes, please add this discussion and your interpretation. Same goes to the relevance of the effects seen in the microscope vs under photocatalytic conditions.

Response) Thank you very much for your valuable comment. Controlling the excitation intensity is critical in triggering crystal damaging reactions. Thus, quantifying the precise threshold excitation intensity is essential for rationalizing and optimizing photocatalytic systems. However, accurately determining this threshold is challenging. Our proposed crystal damaging reactions occur following phase segregation, which is triggered by charge carrier trapping at defects. Since defect density varies from crystal to crystal, it is difficult to quantify a universal threshold. Nonetheless, we observed that excitation intensities of at least $780 \text{ mW} \cdot \text{cm}^{-2}$ or higher induce prompt photodamaging reactions under the microscope. We have added this information in lines 110-112.

6. Maybe some clarification would be needed here: in the microscopy setup light is used to probe PL, but also light is used to trigger destruction. Are these effects / actions separated? How do the authors ensure that data acquisition takes place under “steady-state” conditions? It is not clear from the text and discussions.

Response) Thank you very much for your valuable comment. As shown in Supplementary Fig. 11, the perovskite crystal was damaged following phase segregation, indicating that their processing timescales were different. This allows us to observe and analyze their behavior separately using single-particle spectroscopy. We have added this information in the caption of Supplementary Fig. 11.

Responses to the Comments Given by Reviewer #3:

In this study, the authors explore the photoinduced destruction and healing, over multiple cycles, of a perovskite material using MAPbBr_{2.8}IO₃ as a model system. The authors present a well-informed hypothesis based on their observations supported by several results of their characterization techniques, specifically, single particle spectroscopy. The authors propose the Le-chatelier's principle as the driving force towards this self healing ability of the material. Finally, the authors demonstrate the use of such materials towards H₂ production from HX (X = Br, I), both in the presence and absence of visible light – thereby suggesting a photosystem working based on the day-night cycle on earth. While the authors also attempt to rule out other factors using a set of reference experiments, I believe there are a few important questions that need to be addressed that can further improve the clarity of their message overall.

Response) Thank you very much for your recommendation following the revisions and your thoughtful consideration of our work. Our paper was significantly improved by your helpful advice. We have revised our manuscript in line with your comments.

Major comments:

Questions regarding the self-healing mechanism and experiments

Q1: While the authors propose the Le-chatelier's principle as the reason driving the self healing process, the perovskite MAPbBr_{2.8}IO₂ only demonstrates the self healing property in the absence of light (as mentioned in Lines 113 and 114), as opposed to a simultaneous healing process. Can the authors comment why such a behaviour is observed? Would they not expect a more immediate self-healing process due to the changes induced, according to the Le-chatelier's principle?

Response) This is a highly valuable comment. The immediate self-healing process you mentioned and crystal damaging reactions likely occur simultaneously, especially during photocatalytic reactions. To distinguish these processes, we employed single-particle spectroscopy techniques. As shown in Fig. 2g–k, photodamaging reactions occurred within 300 seconds, while self-healing reactions were completed 1800 seconds after stopping irradiation. Thus, we were able to investigate these two processes separately. To assess immediate self-healing under light, further studies are required; however, this is beyond the scope of the present study. Thank you very much again for your valuable comment.

Q2: Does the overall Br / I ratio remain the same after multiple cycles of healing, or is that changed depending on the surrounding solution that only contains one of the anions available?

Response) Thank you very much for your valuable comment. We measured the photoluminescence (PL) spectra before photodamaging, after photodamaging, and after self-healing reactions. These spectra have been added as Supplementary Fig. 15 and referenced in lines 243-247. In this figure, the PL wavelength, which reflects the halide composition of perovskites, remained unchanged after the self-healing process. Hence, the photodamaging and self-healing processes do not affect the Br/I ratio of the crystal.

Q3: The authors suggest the destruction to be surface sensitive, however, ESI Figure 11 shows the phase segregation in the whole of the crystal? why is that? Is the crystal very small compared to light penetration depth?

Response) Thank you very much for your valuable comment. As you mentioned, in our microscopic experiments, we observe PL only from the near-surface region (~120 nm) due to the limited penetration depth of light. We have added this information in lines 187-188. In Supplementary Fig. 11, although it appears that phase segregation occurred throughout the entire crystal, the actual segregation was confined to the surface region, and the observed PL originated from the surface. We have added a comment on this point in the caption of Supplementary Fig. 11.

Questions regarding photocatalysis

Q4: While there is plenty of text in the manuscript on the proposed H₂ production mechanism, I would recommend the authors to present a few reaction equations showing the oxidation and reduction reactions involved giving the reader a clear idea of the role of the photogenerated electrons and holes in the process.

Response) Thank you very much for your valuable comment. We have added the following reaction equations in lines 261-266.

Q5: While the authors show the band diagrams of the pure Br₃ and I₃ perovskites to be capable of the hydrogen evolution reaction (Figure S14), the observed CB values are far below (more negative) vs the mixed halide composites. Why in this case do Br₃ and I₃ compounds underperform in HER?

Response) Thank you very much for your valuable comment. As described in lines 278-283, one possible reason is light-induced phase segregation. Segregated iodide-rich domains are generated near the surface region. As shown in Supplementary Fig. 6, iodide-rich domains have a narrower band gap compared to mixed-halide and bromide-rich domains, leading to more efficient charge transfer to the surface region. As a result, high HER activity was observed in mixed-halide perovskites. Another possibility is related to the crystal destruction process. Phase segregation induces crystal destruction and hydrogen evolution, as shown in Eq (2). This process did not proceed significantly in bromide and iodide perovskites, which resulted in their lower hydrogen evolution activity.

Q6: The photocatalytic performance of the best performing perovskite MAPbBr_{2.8}I_{0.2} seems to be around 2.5 μmol of H₂ produced in the first hour (Figure 4b). However, in the reference photocatalytic experiment results shown in Supplementary Figure 16, the material seems to only produce ~1.2 μmol of H₂ in the first hour. Could the authors comment on such a fluctuation observed in the initial photocatalytic performance?

Response) Thank you very much for your valuable comment. We evaluated the hydrogen production activity shown in Fig. 4b and Supplementary Fig. 18 under the same conditions. However, as you mentioned, their activity differed. This is likely because properties such as crystallinity and defect density vary depending on the sample synthesis lot, even when synthesized using the same method. To achieve comparable hydrogen production activity regardless of sample lot, large-scale synthesis could be a possible solution.

Q7: while the 1.3/1.7 shows red shift in PL the 0.3/2.7 does not. Could the authors give any comment as to why that happens?

Response) Thank you very much for your valuable comment. The threshold halide composition for phase segregation has been reported as MAPbBr_{0.6}I_{2.4}. If the ratio of bromide anions falls below 0.6, phase segregation does not occur. This can be explained by thermodynamic stability. We have added this comment in the caption of Supplementary Fig. 7.

Minor comments:

Q8: It seems that in ESI Fig 8 and 9 the intensity of the peak increases over illumination time for the I-perovskite and not really for the Br perovskite. Neither does the intensity of Br perovskite go down upon destruction – why is that?

Response) Thank you very much for your valuable comment. It has been reported that semiconductor materials, including perovskites, exhibit an increase in PL intensity under photoirradiation, a phenomenon known as photo-activation or light-curing. In MAPbI₃, photo-activation is attributed to the annihilation of Frenkel defects under photoirradiation. Additionally, it has been reported that the defect formation energy of MAPbBr₃ is higher than that of MAPbI₃, indicating that the defect density in MAPbBr₃ is lower than in MAPbI₃. Thus, significant photo-activation was not observed in MAPbBr₃. The reason why PL from bromide perovskites did not decrease is that crystal destruction primarily occurs in iodide-rich domains, while bromide-rich domains remain relatively intact, as shown in Supplementary Fig. 11. We have added these discussions in the captions of Supplementary Figs. 8 and 9.

Q9: Figure S9: The sliced PL at 241 s shows the segregation but, in the time-plot, the intensity of 700 nm band at 300 s seems to have gone back to 0. Could the authors add the slice at 300 s as well, and comment on this dataset (also relevant to Line 161-164)

Response) Thank you very much for your valuable comment. As shown in Fig. 3f and Supplementary Fig. 11, crystal destruction occurs in iodide-rich domains, which emit ~700 nm PL. Therefore, ~700 nm PL intensity drops to zero at 300 seconds. We have added this comment in the caption of Supplementary Fig. 9.

Q10: In the discussion relating to phase segregation as the reason behind crystal destruction, in Line 186, they say that the holes are trapped by the halide ions in the solution and therefore the electrons (excess) reduce the perovskite and destroy it. Would this also occur if the perovskite would not be immersed in halides saturated solution (in this case there is no halide ions to trap holes)? This would mean that both healing AND DESTRUCTION would require this specific conditions. Please clarify

Response) Thank you very much for your valuable comment. Perovskites can stably exist in solution only when a sufficient concentration of halide anions is present (e.g., [I⁻] > 3.16 M for MAPbI₃). Thus, it is impossible to observe perovskites in the absence of halide anions in solution. Instead, we observed perovskites in air, where no halide anions were present. Under this condition,

significant crystal destruction was not observed. This indicates that hole trapping by halide anions is required or facilitates crystal destruction. We have added this discussion in lines 191-192.

Q11: Line 204, clarify which Pb oxidation state is mentioned.

Response) Thank you very much for your valuable comment. We have mentioned this.

Q12: Fig 4b missing x axis legend

Response) Thank you very much for your valuable comment. We have made the correction.